# *Wolbachia* Infection through Hybridization to Enhance an Incompatible Insect Technique-Based Suppression of *Aedes albopictus* in Eastern Spain

**DOI:** 10.3390/insects15030206

**Published:** 2024-03-20

**Authors:** Maria Cholvi, María Trelis, Rubén Bueno-Marí, Messaoud Khoubbane, Rosario Gil, Antonio Marcilla, Riccardo Moretti

**Affiliations:** 1Area of Parasitology, Department of Pharmacy and Pharmaceutical Technology and Parasitology, Faculty of Pharmacy, Universitat de València, 46100 Valencia, Spain; cholvimaria@gmail.com (M.C.); maria.trelis@uv.es (M.T.); messaoud.khoubbane@uv.es (M.K.); antonio.marcilla@uv.es (A.M.); 2Joint Research Unit in Endocrinology, Nutrition and Clinical Dietetics, La Fe Health Research Institute, 46026 Valencia, Spain; 3Research and Development Department, Lokímica Laboratories, 46980 Paterna, Spain; ruben.bueno@rentokil-initial.com; 4Center of Excellence in Vector Control for Europe, Rentokil Initial, 46980 Paterna, Spain; 5Institute of Integrative Systems Biology (I2SYSBIO), Universitat de València/CSIC, 46980 Paterna, Spain; rosario.gil@uv.es; 6Biotechnology and Agroindustry Department (SSPT-BIOAG-SOQUAS), ENEA (Italian National Agency for Biotechnology, Energy and Sustainable Economic Development), 00123 Rome, Italy

**Keywords:** Asian tiger mosquito, genetic control, *Wolbachia*, cytoplasmic incompatibility, population suppression, hybridization, genotyping, arboviruses

## Abstract

**Simple Summary:**

*Wolbachia* bacteria occur naturally as symbionts of many insect species and are responsible for various phenomena that modify the hosts’ reproductive biology. Among them, cytoplasmic incompatibility (CI) refers to the sterility of eggs produced by crosses between infected males and females that are uninfected or infected by a non-compatible strain of these bacteria. CI can be exploited for vector control by establishing an opportune *Wolbachia* infection in a laboratory population of a target insect species and then releasing the infected males into the environment as sterilizing agents. In the present work, a suitable *Wolbachia* strain was introduced into a Spanish population of the Asian tiger mosquito, *Aedes albopictus*, through hybridization with the laboratory line, AR*w*P, already tested as an efficient control tool against this vector. The obtained hybrids were compared with the AR*w*P to ascertain the effects derived from transferring the infection to a different *Ae. albopictus* population. No significant differences between lines were found regarding survival, female fecundity, and egg fertility. Importantly, the eggs produced by crosses between males of the hybrid lines and unmodified wild females were 99.9% sterile. This result encourages further studies to explore the feasibility of a *Wolbachia*-based control program against the Asian tiger mosquito in Spain.

**Abstract:**

The emergence of insecticide resistance in arbovirus vectors is putting the focus on the development of new strategies for control. In this regard, the exploitation of *Wolbachia* endosymbionts is receiving increasing attention due to its demonstrated effectiveness in reducing the vectorial capacity of *Aedes* mosquitoes. Here, we describe the establishment of a naïve *Wolbachia* infection in a wild *Aedes albopictus* population of eastern Spain through a hybridization approach to obtain males capable of sterilizing wild females. The obtained lines were compared with the *Wolbachia* donor, *Ae. albopictus* AR*w*P, previously artificially infected with *Wolbachia w*Pip, regarding immature and adult survival, female fecundity, egg fertility, and level of induced sterility. Our results did not show significant differences between lines in any of the biological parameters analyzed, indicating the full suitability of the hybrids to be used as a control tool against *Ae. albopictus*. In particular, hybrid males induced 99.9% sterility in the eggs of wild females without the need for any preliminary treatment. Being harmless to non-target organisms and the environment, the use of this bacterium for the control of *Ae. albopictus* deserves further exploration. This is especially relevant in areas such as eastern Spain, where this mosquito species has recently spread and may represent a serious threat due to its competence as a vector for dengue, chikungunya, and Zika viruses.

## 1. Introduction

Strategies aimed at controlling mosquito-borne diseases still seem far from the goal of significantly and steadily reducing infections, deaths, and economic losses due to the costs of prevention, control treatments, and cures [1,2]. Instead, also due to climate warming, urbanization, and connectivity, key vectors keep on expanding their geographical distribution, especially in temperate climate areas, leading to an increase in the global population at risk of arboviral diseases [3]. In the current context of global change, it is a priority to focus on the development of best practices for mosquito control while seeking to avoid the consequences of the further spread of infectious disease vectors [4]. However, this objective should be reconciled with the need to meet the Sustainable Development Goals (SDGs), with particular attention to the protection of the environment and biodiversity [5].

The development of vaccines and prophylaxis drugs targeting arboviruses, along with prevention measures and community involvement, have certainly helped in reducing health-related risks [4,6], but mosquito control methods have often shown only temporary success [7,8]. This is mainly due to the remarkable capacity of mosquitoes to adapt to a changing environment [9] and to develop insecticide resistance [10,11]. Also, insecticide spraying is often unable to reach cryptic habitats of immature and adult mosquitoes, especially in urban areas [12]. Insecticide resistance and the negative side effects of insecticides on non-target organisms and the environment make the integration of chemical control with other vector suppression strategies a priority [4,7,13,14]. The lack of effective vaccines or drugs against certain vector-borne pathogens further stresses this need [15,16].

In search of effective and environmentally friendly methods to fight disease vectors, autocidal control strategies are considered a promising option to be integrated with other control measures [17,18]. These strategies are based on the continuous release of large numbers of sterile males belonging to the target species to reduce the chance of wild females reproducing and, consequently, leading to a gradual decline in the resident vector population. Male sterilization can be achieved through the administration of ionizing radiation (sterile insect technique, SIT) or sterilizing chemicals, or by taking advantage of a phenomenon of reproductive incompatibility induced by bacteria of the genus *Wolbachia* (incompatible insect technique, IIT). Genetic engineering can also pursue the same goal, for example, obtaining sterile males through the CRISPR/Cas9 method [19], or producing laboratory lines carrying a late-acting dominant lethal gene that is transmitted to the offspring of wild females by mating (release of insects carrying a dominant lethal, RIDL; [20,21]).

Among the aforementioned strategies, IIT is receiving increasing attention, because the incompatible males’ ability to completely sterilize wild females is accompanied by a full preservation of male fitness since they have not undergone any pre-release sterilizing treatment or genetic modification [22,23]. This is a clear advantage over other genetic control approaches in terms of achieving high effectiveness but also safety and sustainability, and facilitating public acceptance [24,25].

*Wolbachia* are endosymbionts of nematodes and arthropods; among the latter, they are estimated to infect approximately 40% of all species [26]. Although the interaction with the hosts is generally mutualistic, in many cases, *Wolbachia* are responsible for profound alterations in the hosts’ reproductive biology and physiology, which favor the spread of the infection even when only a small percentage of the individuals are initially infected [27,28,29]. IIT takes advantage of one of these physiological alterations, the induction of the phenomenon of cytoplasmic incompatibility (CI). CI can be described basically as a failure in the development of the embryo when a *Wolbachia*-infected male inseminates either an uninfected female or a female infected by an incompatible *Wolbachia* strain [30]. Based on the specific features of the known *Wolbachia* strains, various infections have been established artificially in *Aedes albopictus* and *Aedes aegypti* to exploit CI as a suppression tool [31,32,33,34,35,36]. IIT has already demonstrated its effectiveness in suppressing both *Ae. albopictus* and *Ae. aegypti* in different areas of the world, either alone [31,36,37,38,39,40,41] or in combination with SIT [42,43], gradually increasing the record of successes achieved.

Given a target vector species and a target area, the first step in an IIT-based control program is the generation of a suitable *Wolbachia*-infected laboratory line of the target species whose males can fully sterilize wild females. A new association of *Wolbachia* with a host is generally artificially generated by embryonic or adult transinfection by microinjection [44]. However, other populations of the same species can be subsequently infected by hybridization (i.e., crossing wild males with artificially infected females of the same species). This process can be easy when unidirectional CI patterns occur, as in the case of *Ae. aegypti* [31], but it is also feasible in the case of bidirectional CI patterns that are not full. This is the case with *Ae. albopictus* AR*w*P, a *Wolbachia*-transinfected line harboring *Wolbachia w*Pip [32], and wild populations (*w*AlbA-/*w*AlbB-infected) of this mosquito species. In fact, wild *Ae. albopictus* males have been found to become partially fertile with aging when they inseminate *w*Pip-infected *Ae. albopictus* females. This phenomenon is due to the age-dependent depletion of *w*AlbA in males, while *w*AlbB alone is unable to induce a full CI pattern in the crosses with wPip-infected females [45].

Once a laboratory line capable of producing incompatible males has been obtained, further studies are necessary before open-field releases. In fact, the effectiveness of the strategy depends not only on the number of released males but also on the level of induced CI and their quality in terms of survival and mating competitiveness [46,47,48]. The genetic background has a demonstrated effect on certain characteristics of the host–*Wolbachia* association [49,50] and, additionally, artificial rearing conditions may gradually induce selection for biological traits that can be advantageous in mass-rearing settings but might not be fully appropriate in the open field [51]. Moreover, different environmental contexts and different target wild populations may require specific adjustments to maximize results. In this regard, it has been demonstrated that matching the local population genetics, especially its insecticide resistance background, was critical to achieving a successful invasion in the case of releases of *Wolbachia*-infected *Ae. aegypti* aiming at the replacement of the wild population [52]. This preliminary step can also reasonably fit the IIT strategy because sterile males with the same genetic background as the target population are more likely to have also acquired alleles conferring resistance to certain insecticides or adaptability to the local climate.

In the present work, we report the results of a hybridization process aiming at introducing *w*Pip *Wolbachia* from *Ae. albopictus* AR*w*P into the genetic background of an *Ae. albopictus* population from Barcelona (Spain). The success of this procedure was assessed by PCR targeting *Wolbachia* and mtDNA-specific genes. Results of the characterization of hybrid lines to evaluate their suitability for mass rearing and as a genetic control tool are also reported.

These studies are part of a research project aiming at exploring the feasibility of a *Wolbachia*-based control program against *Ae. albopictus* in eastern Spain. In 2004, this vector was recorded for the first time in the country [53]; twenty years later, the species is now established in 40 provinces of Spain and has become a serious threat to human welfare and health [54]. Indeed, the impact of this invasion in economic and public health terms has resulted in the investment of hundreds of thousands of euros in mosquito control programs. Despite this, the occurrence of several autochthonous dengue cases in Mediterranean areas of Spain [54] highlights the urgency of new control measures and renewed coordinated efforts to prevent serious outbreaks of arboviral diseases.

## 2. Materials and Methods

### 2.1. Ae. albopictus Lines and Their Maintenance

The line, B_N_, was derived from a wild *Ae. albopictus* population from Barcelona (Spain), kindly provided by Dr. Mara Moreno-Gómez (Henkel Ibérica, S.A., Barcelona, Spain) and maintained in laboratory conditions at Universitat de València since 2020. As commonly found worldwide in *Ae. albopictus*, this population naturally harbors *Wolbachia w*AlbA and *w*AlbB, but some wild individuals lacking one or both of these two strains have been previously reported in eastern Spain [55]. The line, B_A_, is a *Wolbachia*-free *Ae. albopictus* population obtained at the Universitat de València by curing the natural *Wolbachia* infection of B_N_, according to the methods described by Dobson and Rattanadechakul [56]. The *Ae. albopictus* AR*w*P line was established in 2008 at the Casaccia Research Center of the National Agency for New Technologies, Energy and Sustainable Economic Development (CR ENEA Casaccia, Rome, Italy) through embryonic microinjection of *Wolbachia w*Pip isolated from a wild population of *Culex pipiens molestus* into a wild local population that was cured of natural *Wolbachia* infections [32]. This line is characterized by a bidirectional CI pattern in crosses with wild *Ae. albopictus* [22]. A specific sub-population of AR*w*P, called AR*w*P_L_, was kept isolated from the rest of the colony to perform all the experiments described here. The three populations were maintained at the CR ENEA Casaccia, where all the experiments involving live insects were performed.

Unless stated differently, the larvae belonging to the above populations were reared inside 1L larval trays at a density of 1 larva/mL, provided with increasing doses (0.2, 0.4, 0.6, and 0.8 mg × larva × day) of a liquid diet consisting of 50% tuna meal, 36% bovine liver powder, and 14% brewer’s yeast (according to the IAEA-BY diet but without the addition of vitamins) [39]. Adults were maintained inside 30 × 30 × 30 cm cages at a temperature of 27.0 ± 0.5 °C, relative humidity (RH) of 70.0% ± 10.0%, and 14:10 h light/dark cycle, and they were supplied with cotton balls soaked in 10% sucrose *ad libitum*. Blood meals were provided via anesthetized mice in agreement with the Bioethics Committee for Animal Experimentation in Biomedical Research and in accordance with procedures approved by the ENEA Bioethical Committee according to EU Directive 2010/63/EU. The mice belonged to a colony housed at CR ENEA Casaccia, and they were maintained for experimentation based on authorization no. 80/2017-PR released (on 2 February 2017) by the Italian Ministry of Health.

### 2.2. Hybridization Protocol

Two hybrid lines were established in parallel. The AR*w*P_BA_ line was developed by crossing AR*w*P_L_ females with 3 ± 1-day-old B_A_ males. In this type of crossing, a similar percentage of egg hatching was expected as in that of AR*w*P_L_ male × AR*w*P_L_ female crosses, because uninfected males are fully compatible with *Wolbachia*-infected females. The sex of the progeny was determined to allow for the use of the hybrid females in a subsequent cross with B_A_ males. This procedure was repeated for 5 generations (G_1_–G_5_).

Crosses between AR*w*P_L_ females and B_N_ males were also performed for 5 generations. In this case, 3 ± 1-day-old AR*w*P_L_ females were mated with 18 ± 1-day-old B_N_ males to exploit the male age-related reduction of the CI level in wild male × *w*Pip-infected female crosses [47]. The mean level of egg fertility in AR*w*P_L_ females × aged B_N_ males was measured. Even the reduced egg fertility was expected to be enough to obtain a sufficient number of female larvae to be used for the subsequent hybridization cycles.

After 5 generations of backcrossing, the AR*w*P_BA_ and AR*w*P_BN_ lines were expected to have about 97% of the genetic background of the B_A_ and B_N_ lines, respectively, while harboring *Wolbachia w*Pip from AR*w*P_L_.

### 2.3. Monitoring of the Establishment and Maternal Transmission of the Infection

The establishment of the *w*Pip infection was monitored during the hybridization process by randomly sampling 10 females in each generation (G_1_–G_4_) and identifying the percentage of *w*Pip-infected individuals by polymerase chain reaction (PCR) analysis. At G_5_, the maternal transmission of the infection was similarly measured by counting the *w*Pip-positive individuals out of 20 females born to mothers previously tested positive for *w*Pip.

DNA was extracted from individual mosquitoes by dissecting and homogenizing their abdomens in 100 μL of STE buffer with 0.4 mg/mL of proteinase K [57]. A *w*Pip-specific primer pair, targeting the *wsp* gene [58], was designed and used to recognize *Wolbachia w*Pip: *w*Pip-F bis: 5′-GTTAGTGGTGCAACATTTACTCC-3′; *w*Pip-R: 5′-AATAACGAGCACCAGCAAAGAGT-3′. The PCR cycling procedure was 94 °C for 5 min followed by 35 cycles of 94 °C for 30 s, 56 °C for 30 s, 72 °C for 40 s, and a single final step at 72 °C for 10 min. Amplicons were electrophoresed on 1.5% agarose gels, stained with ethidium bromide (1 μg/mL), and visualized under ultraviolet light. The *w*Pip-negative samples were then PCR-tested to ascertain the absence of *Wolbachia* or, eventually, the presence of *Wolbachia w*AlbA and *w*AlbB by using primers and PCR protocols specific to these strains [56].

### 2.4. Sequencing of the COI Gene in Hybrid Populations

To confirm the introgression of the mtDNA together with *Wolbachia* in comparison with the original wild-type line, B_N_, *Ae. albopictus* lines B_A_, B_N_, AR*w*P_L_, AR*w*P_BA_, and AR*w*P_BN_ were also characterized by targeting the cytochrome oxidase I (COI) gene as mitochondrial DNA (mtDNA) marker, as previously performed in similar studies [59]. DNA from whole adult female mosquitoes (5 individuals per line) was extracted with the commercial DNeasy Blood + Tissue DNA Extraction Kit, Qiagen (Hilden, Germany) following the manufacturer’s instructions. The samples were analyzed by PCR using the primers LCOI490 (5′-GGTCAACAAATCATAAAGATATTGG-3′) and HCO2198 (5′-TAAACTTCAGGGTGACCAAAAAATCA-3′), as described by Lucati et al. [59]: 5 min at 95 °C, followed by 5 cycles of 60 s at 95 °C, 90 s at 45 °C, 45 s at 72 °C, and then 30 cycles of 45 s at 95 °C, 45 s at 50 °C, and 45 s at 72 °C, concluding with a last step of 7 min at 72 °C. The PCR products were checked by 2% agarose gel electrophoresis, stained with SafeView Classic stain (NBS Biologicals, Huntingdon, Cambridgeshire, UK), and visualized under UV light. Positive samples were sent to the sequencing facility of the Universitat de València (Servicio Central de Soporte a la Investigación Experimental, SCSIE) for the purification and sequencing of the samples. The quality of sequencing products was assessed by manual inspection of each chromatogram. The sequences were aligned using CLUSTALW2 [60] with default settings and compared with sequences in the databases by using BLASTN (http://www.ncbi.nlm.nih.gov/BLAST accessed on 1 March 2024). The DNA sequences determined in this study have been deposited in GenBank (see Accession Numbers in Appendix A).

### 2.5. Determination of Fitness Parameters and Induced CI Levels of the Hybrid Lines

At G_5_, AR*w*P_BA_, and AR*w*P_BN_ were compared with AR*w*P_L_ with regard to immature survival until the adult stage, adult survival, female fecundity, and egg fertility. The CI level induced by AR*w*P_BA_ and AR*w*P_BN_ males when crossed with B_N_ females was also measured. Specifically, three larval trays with 100 L1 larvae in 100 mL of water and feeding were prepared per line, as above described. The obtained adults were counted to measure the percentage of larvae that survived to the adult stage. Three cages per line were then populated by 20:20 males:females. Adult survival was measured by counting and removing from the cages the dead individuals three times per week (Monday, Wednesday, and Friday) over 40 days, taking into account that the mean adult survival in nature is not expected to exceed about 20 days [61]. When 1-week-old females were provided with a blood meal and produced eggs, they were collected using oviposition cups half-filled with rainwater and provided with germination paper lining the inside. Eggs were allowed to dry, counted, and then hatched to measure female fecundity (eggs/female) and egg fertility (egg hatch rate).

For CI-level studies, 10 males belonging to the two hybrid lines were crossed with 10 B_N_ females. After blood-feeding, eggs were hatched to measure their fertility. B_N_ female × B_N_ male crosses were also conducted as a control. The experiment was repeated 3 times.

### 2.6. Data Analysis

Results were expressed as mean ± standard error of the mean (SEM), and the arcsine square root transformation was applied to analyze proportional data. The Brown–Forsythe and the Shapiro–Wilk tests were performed to assess equality of variances and normality, respectively.

The survival curves of AR*w*P_BA_, AR*w*P_BN_, and AR*w*P adults were compared using the Kaplan–Meier method and the log-rank (Mantel–Cox) test. A one-way analysis of variance (ANOVA) was used to compare female fecundity, egg fertility, and immature survival between lines. An ANOVA was also used to compare the level of induced CI in the crosses of B_N_ females with males of AR*w*P_BN_, AR*w*P_BA_, and B_N_. In the case of significant *p* values the Holm–Sidak test was performed as a post hoc test.

Statistical analysis was performed using the Sigma-Stat software (Sigma-Stat 4.0, Systat Software Inc., San José, CA, USA), with the level of significance set at *p* < 0.05.

## 3. Results

### 3.1. Establishment and Maternal Transmission of the Infection

As confirmed by PCR analysis, both hybridization protocols succeeded in establishing *Wolbachia w*Pip in *Ae. albopictus* B_A_ and B_N_, even though the average fertility in crosses between B_N_ males aged 18 ± 1 days and AR*w*P females was only 4.70 ± 0.41%. Nevertheless, not all AR*w*P_BA_ and AR*w*P_BN_ individuals were found to be infected with *Wolbachia w*Pip (Figure 1). In the AR*w*P_BA_, *w*Pip infection ranged from 80 to 100% in females, and, at G_5_, two out of twenty individuals obtained from *w*Pip-positive mothers were found uninfected, indicating an imperfect maternal transmission of the infection. Instead, in the AR*w*P_BN_, the infection by *Wolbachia w*Pip encountered 100% fixation by G_3_ and continued up to G5. All the individuals that were negative to *w*Pip were found also negative to *w*AlbA and *w*AlbB.

### 3.2. Comparison of the COI Gene Sequences between Ae. albopictus Lines

The mtDNA alignment of the COI gene was obtained for five sequences of about 500 bp per mosquito line, and no evidence of genetic diversity, regarding haplotypes or even single nucleotides, was detected (Appendix A). Consequently, at the level of this gene, it was not possible to highlight any genetic dissimilarity between *Ae. albopictus* lines B_A_, B_N_, AR*w*P, AR*w*P_BA_, and AR*w*P_BN_.

### 3.3. Fitness and CI Levels of Hybrid Lines Compared to ARwP

The immature survival rate did not differ significantly between *Ae. albopictus* AR*w*P_L_, AR*w*P_BA_, and AR*w*P_BN_ (F_2,6_ = 2.08; *p* = 0.21; Figure 2) and ranged from 81 to 90%.

Adult survival was not found to significantly differ between *Ae. albopictus* AR*w*P_L_, AR*w*P_BA_, and AR*w*P_BN,_ and this result was confirmed in both sexes (Figure 3). Mean survival ranged from 31 to 32 days in males of the three *Ae. albopictus* lines (*p* = 0.724, log-rank test) while it was higher in females, where it ranged from 36 to 38 days in the three tested lines (*p* = 0.519, log-rank test).

Both mean female fecundity (F_2,6_ = 2.72; *p* = 0.14; Figure 4, left) and egg fertility (F_2,6_ = 0.14; *p* = 0.88; Figure 4, right) were found to not significantly differ between the three *Wolbachia w*Pip-infected lines.

As shown in Figure 5, the AR*w*P_BN_ and AR*w*P_BA_ males acquired an almost full level of induced CI when crossed with BN females (about 99.9%), and the differences in egg fertility compared to the control were significant (F_2,12_ = 1010.83; *p* < 0.05; Figure 5). The differences between the two hybrid lines were not significant (F_1,8_ = 0.03; *p* = 0.86).

## 4. Discussion

Newly established associations between *Wolbachia* and host species may go through a phase of unstable equilibrium due to genetic conflict between the two organisms [27]. In certain cases, the host can be unsuitable for these symbionts [62], or the infection is temporarily successful but is lost within a few generations [63]. In other cases, *Wolbachia* becomes established and may be apparently neutral to the host [64], may negatively affect its fitness [65], or enhance it instead [66], but several generations may be necessary for the stabilization of the symbiosis through selection [47,67].

Negative effects on host fitness due to *Wolbachia* have been previously reported to occur, especially in the first generations after the establishment of the infection, and may affect immature mortality, adult longevity, female fecundity, and egg fertility [32,66,68,69]. *Wolbachia* can alter these traits through direct interaction with host cytoplasmic and nuclear components [70] but also indirectly because of the alteration of the microbial community, which may affect the expression of genes involved in metabolism [71].

In the case of the *Ae. albopictus* lines tested here, AR*w*P_BN_ and AR*w*P_BA_, the naïve infection by *Wolbachia w*Pip was found to rapidly establish even though not all individuals tested positive for the presence of the bacteria. This result could be partly due to the lack of sensitivity of standard PCR when *Wolbachia* density is very low and only one *Wolbachia* target gene is used [45,72,73], but it may also highlight a gradual process of coadaptation by selection between *Wolbachia* and the new genetic background of the two lines. Indeed, genotypes not favoring the establishment and full maternal transmission of the infection are expected to be counter-selected and gradually eliminated due to the CI phenomenon [74]. The complete fixation of the infection by G_3_ in AR*w*P_BN_ seems to attest to a fast accomplishment of this process in this line, but AR*w*P_BA_ can also reasonably be expected to reach this goal in a few more generations [40]. Nonetheless, the almost full level of CI induced by G_5_ AR*w*P_BA_ males could indicate that the proportion of *Wolbachia w*Pip-infected individuals was already closer to 100% than expected based on the PCR results.

The first documented introduction of *Wolbachia w*Pip in *Ae. albopictus* (in replacement of the double *w*AlbA–*w*AlbB natural infection) caused an initial reduction in female fecundity and egg fertility compared to the wild and aposymbiotic controls, and this fitness cost persisted for several generations after the establishment of the symbiotic association [32]. These effects gradually faded over time, due to selection and outcrossing. In fact, at present, AR*w*P fitness does not differ significantly from that of wild individuals characterized by the same genetic background [40]. In contrast, our results demonstrated that *Wolbachia w*Pip was not responsible for any evident fitness effect in the two hybrid lines compared to AR*w*P. The female fecundity and egg fertility of hybrid lines were also found to be not significantly different compared to the B_N_ line, demonstrating a not significant fitness effect related to the *Wolbachia* strain regarding these parameters (Appendix A).

The absence of any significant difference between AR*w*P_L_, AR*w*P_BN_, and AR*w*P_BA_ with regard to immature and adult survival, female fecundity, and egg fertility may suggest coadaptation of *Wolbachia w*Pip with the host *Ae. albopictus* over 15 years of coevolution. This phenomenon has been highlighted before in other systems and can also be very fast due to selection [75,76].

Nevertheless, it is also possible that AR*w*P (derived from an Italian wild population caught in an area north of Rome [32]) and B_N_ (established from individuals collected in Barcelona), even if collected in geographic areas thousands of kilometers apart, might not be so distant from a genetic point of view. The COI gene did not show any significant difference between the mtDNA of the two populations, but this result is not surprising due to the reduced diversity of *Ae. albopictus* for COI [59,77]. Therefore, the hypothesis of genetic similarity between AR*w*P and B_N_ should be ascertained by including in the genotyping also nuclear genes, microsatellites, or single nucleotide polymorphisms (SNPs) that have a higher resolution power [59,78]. If confirmed, this close relationship could suggest the occurrence of recent exchanges of *Ae. albopictus* individuals between the two areas. However, this kind of effort goes beyond the need for a research program aimed primarily at evaluating the feasibility of an IIT-based control program against *Ae. albopictus* in eastern Spain.

As a first step in such a program, the reported hybridization process allowed the production of two *Ae. albopictus* lines, AR*w*P_BN_ and AR*w*P_BA_, that: (i) have males capable of fully sterilizing the wild females without the need for any physical or chemical treatment or genetic manipulation, (ii) are characterized by a common genetic background in an area of eastern Spain, and (iii) show traits fully suitable for their long-term maintenance and mass rearing.

More generally, the reported findings support the suitability of using a hybridization approach to produce an incompatible *Ae. albopictus* line. In fact, establishing a new infection ex novo in a wild population of a vector species through *Wolbachia* transinfection requires specific infrastructures and specialized personnel together with a great deal of time. Indeed, high numbers of microinjection attempts are generally required to succeed (because of the generally low success rate), and the goal can be jeopardized by causes related to the suitability of the host, the stability of the infection, and the possible effects of the novel infection on host fitness [66,76].

Instead, an infection achieved via hybridization has the potential to quicken the process because a specific host–symbiont association can be obtained, not by injecting *Wolbachia*, but by gradually introducing the target genetic background into a *Wolbachia*-infected population that already presents ideal characteristics for IIT programs. Furthermore, the hybridization protocol can be programmed and monitored to obtain the desired result in a defined time by verifying the key biological parameters to predict the adaptability of the new association to mass-rearing conditions and its potential as a suppression tool (in terms of the induced level of CI and male mating competitiveness) [50].

A hybridization approach could also be addressed for the introduction of incompatible males of gene variants conferring resistance to certain insecticides to increase their survival in comparison with wild males when released in a target area subjected to insecticidal treatment. Indeed, concurrent pesticide applications have been suggested as a means to assist a *Wolbachia*-based population replacement if the released individuals carry a higher level of resistance than the resident population [79]. This strategy could also be applied in the case of incompatible male releases in the context of an integrated program of vector control [4,13].

All of these considerations support the efficacy and sustainability of large-scale IIT programs planned to target a defined area with its specific resident vector population and may support a broader application of this strategy to be integrated with other control methods. In the context of the present research, a series of challenges must be addressed to close the gap between laboratory settings and field implementation. Further studies should be conducted to ascertain the male mating competitiveness of hybrid lines in comparison with local wild males under semi-field conditions to evaluate the effectiveness of the strategy [48]. According to other *Wolbachia*-based control programs, a safety assessment study will be also necessary before open field trials [80]. In the case of bi-CI patterns, the safety of releasing incompatible males, even in the presence of a low percentage of residual females, has been already demonstrated because the naïve *Wolbachia* infection is unlikely to spread in the wild population [40]. However, protocols to eliminate female contamination need to be considered [37], and all the possible safety concerns regarding the release of *Wolbachia*-infected mosquitoes must be evaluated. After that, small-scale trials under open field conditions should be implemented to demonstrate the potential of IIT, favor community acceptance, and help to set appropriate release protocols to achieve efficacy and sustainability [38,39]. All these steps are necessary before moving to large-scale programs that imply specific mass-rearing facilities and major investments.

Partnerships with public health organizations, ecological monitoring groups and both public and private mosquito control entities will undoubtedly enhance the scalability and sustainability of this control method.

## 5. Conclusions

Our findings demonstrate that the introgression of the genetic background of a wild population in a *Wolbachia*-infected line capable of producing incompatible males can be a fast option to start to evaluate the feasibility of an IIT program. In the case of *Ae. albopictus*, this objective can be pursued by crossing populations with different *Wolbachia* infections if CI is not full, or by the preliminary establishment of an aposymbiotic line from the wild-type target to facilitate hybridization.

Both the AR*w*P_BA_ and AR*w*P_BN_ hybrid males, derived from these two alternative outcrossing protocols, showed a 99.9% level of induced CI when crossed with wild females sharing the same genetic background and without the need for any sterilizing treatment.

The obtained information can serve as a preliminary step to evaluate the effectiveness of an IIT-based control strategy against *Ae. albopictus* in eastern Spain and help estimate the investments needed for a larger-scale open field program. Furthermore, it can provide the basis for modeling predictions and designing more accurate field experiments aimed at establishing incompatible male release protocols specific to the target context and integrated with other control measures. Such preliminary studies will contribute to improving efficacy and saving costs of the overall control program to achieve positive results but also sustainability in the long term.

## Figures and Tables

**Figure 1 insects-15-00206-f001:**
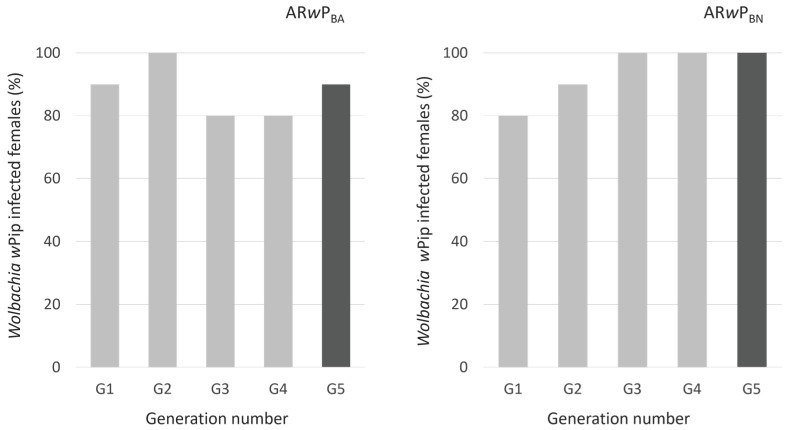
*Wolbachia w*Pip-infected females in *Ae. albopictus* AR*w*P_BA_ and AR*w*P_BN_ over 5 subsequent generations since the first cycle of hybridization. AR*w*P_L_: *Wolbachia w*Pip-infected *Ae. albopictus* from Rome (Italy) [31]; AR*w*P_BN_: wild strain of *Ae. albopictus* from Barcelona (Spain) infected with *Wolbachia w*Pip from AR*w*P_L_ through hybridization; AR*w*P_BA_: aposymbiotic strain of *Ae. albopictus* obtained in Valencia from the wild strain of Barcelona and then infected with *Wolbachia w*Pip from AR*w*P_L_ through hybridization. From G_1_ to G_4_ (light grey), females were randomly sampled among all emerged adults and checked for *w*Pip infection by PCR (n = 10). At G_5_ (dark grey), tested females were born from eggs oviposited by females that had been ascertained to be *w*Pip-positive (n = 20).

**Figure 2 insects-15-00206-f002:**
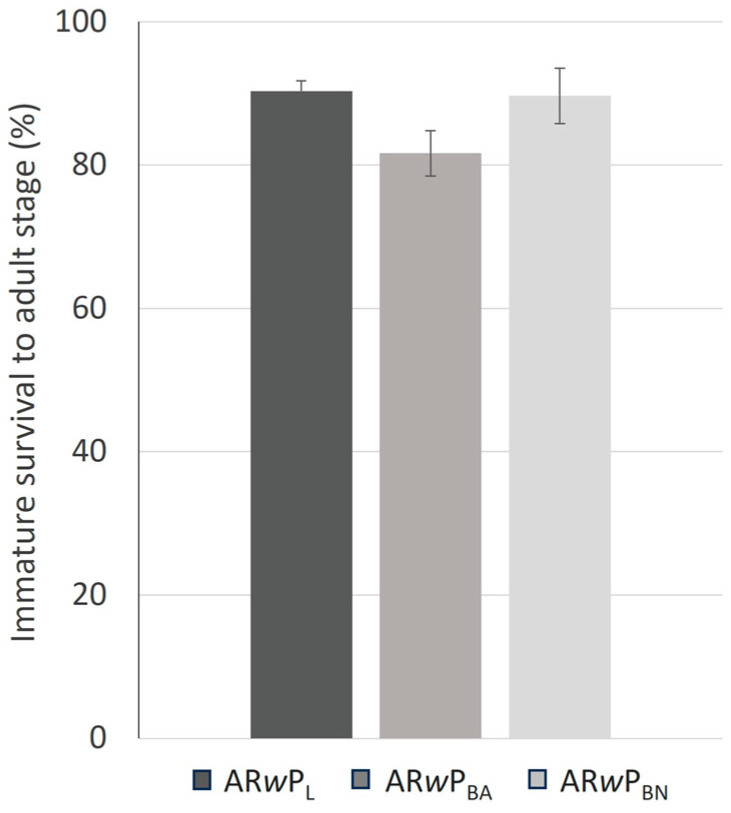
Immature survival of *Ae. albopictus* AR*w*P_L_, AR*w*P_BA_, and AR*w*P_BN_. AR*w*P_L_: *Wolbachia w*Pip-infected *Ae. albopictus* from Rome (Italy) [31]; AR*w*P_BN_: wild strain of *Ae. albopictus* from Barcelona (Spain) infected with *Wolbachia w*Pip from AR*w*P_L_ through hybridization; AR*w*P_BA_: aposymbiotic strain of *Ae. albopictus* obtained in Valencia from the wild strain of Barcelona and then infected with *Wolbachia w*Pip from AR*w*P_L_ through hybridization. Error bars show the standard error of the mean of three repetitions. Differences between lines were not statistically significant by one-way ANOVA.

**Figure 3 insects-15-00206-f003:**
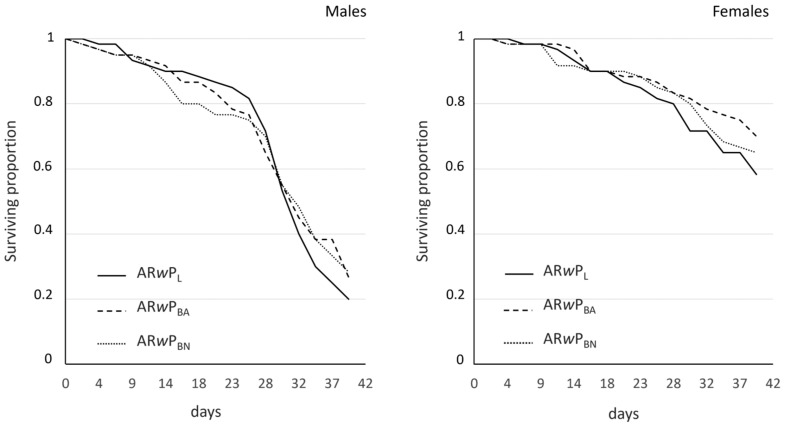
Survival curves of male and female adults belonging to *Ae. albopictus* AR*w*P_L_, AR*w*P_BA_, and AR*w*P_BN_, three lines infected with the same *Wolbachia* strain through hybridization. AR*w*P_L_: *Wolbachia w*Pip-infected *Ae. albopictus* from Rome (Italy) [31]; AR*w*P_BN_: wild strain of *Ae. albopictus* from Barcelona (Spain) infected with *Wolbachia w*Pip from AR*w*P_L_ through hybridization; AR*w*P_BA_: aposymbiotic strain of *Ae. albopictus* obtained in Valencia from the wild strain of Barcelona and then infected with *Wolbachia w*Pip from AR*w*P_L_ through hybridization. In both sexes, differences between lines were not statistically significant by log-rank (Mantel–Cox) test.

**Figure 4 insects-15-00206-f004:**
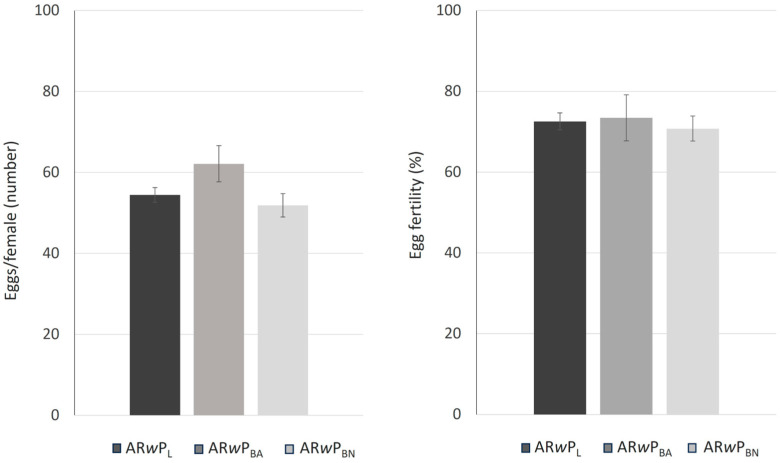
Female fecundity (**left**) and egg fertility (**right**) in *Ae. albopictus* AR*w*P_L_, AR*w*P_BA_, and AR*w*P_BN_. AR*w*P_L_: *Wolbachia w*Pip-infected *Ae. albopictus* from Rome (Italy) [31]; AR*w*P_BN_: wild strain of *Ae. albopictus* from Barcelona (Spain) infected with *Wolbachia w*Pip from AR*w*P_L_ through hybridization; AR*w*P_BA_: aposymbiotic strain of *Ae. albopictus* obtained in Valencia from the wild strain of Barcelona and then infected with *Wolbachia w*Pip from AR*w*P_L_ through hybridization. Error bars show the standard error of the mean (SEM) of three biological replicates, each containing 19–20 fed females. In both cases, values were not significantly different by one-way ANOVA.

**Figure 5 insects-15-00206-f005:**
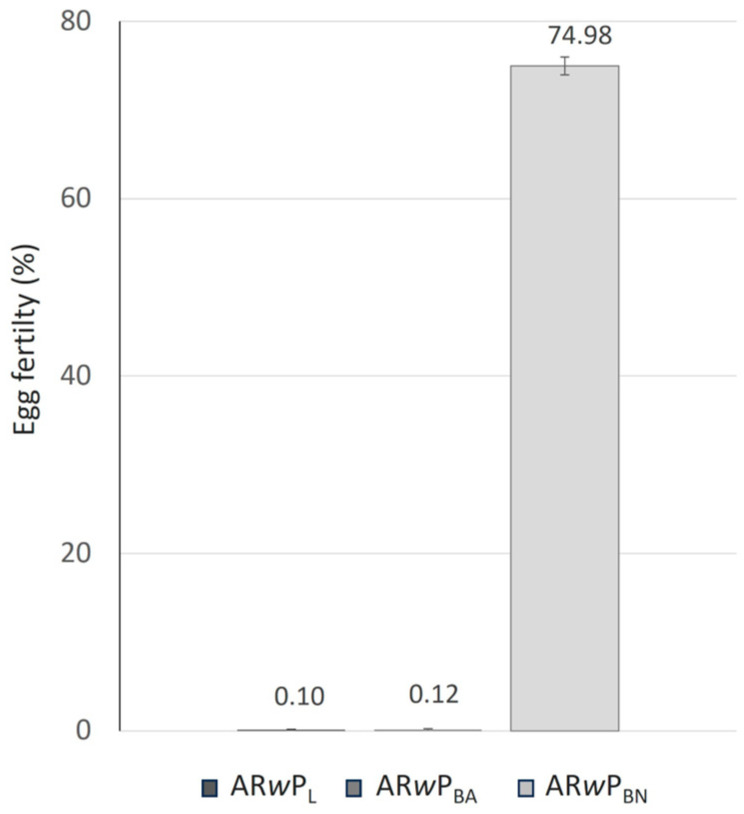
Egg fertility (when *Ae. albopictus* B_N_ females were crossed with *Ae. albopictus* males AR*w*P_BN_, AR*w*P_BA_, and B_N)_. B_N_: wild strain of *Ae. albopictus* from Barcelona colonized in Valencia (Spain); AR*w*P_BN_: wild strain of *Ae. albopictus* from Barcelona (Spain) infected with *Wolbachia w*Pip from AR*w*P_L_ through hybridization; AR*w*P_BA_: aposymbiotic strain of *Ae. albopictus* obtained in Valencia from the wild strain of Barcelona and then infected with *Wolbachia w*Pip from AR*w*P_L_ through hybridization. Error bars show the SEM of three biological replicates. The mean egg fertility was found to differ significantly between crosses by one-way ANOVA.

## Data Availability

All necessary data generated or analyzed during this study are included in this article and its Appendix A. Sequence data generated by this study are available based on the GenBank codes provided in Appendix A. Further inquiries can be directed to the corresponding author.

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
