# Peer review of "Wolbachia Infection through Hybridization to Enhance an Incompatible Insect Technique-Based Suppression of Aedes albopictus in Eastern Spain"

_insects, 2024, doi:10.3390/insects15030206_

Round 1

Reviewer 1 Report

Comments and Suggestions for Authors

The authors of this manuscript describe the introgression of the genetic background of an Ae. albopictus population from Spain with a strain of Ae. albopictus harboring a wPip Wolbachia infection. The methods are sound and appropriate and support the authors conclusions. The manuscript is unfortunately full of improper English language usage and needs to be reviewed by a native English speaker and or an editorial service. The statistical analyses appear appropriate where used.  However, the manuscript only described results that are important for the development of an IIT approach to target localized populations in Spain.  While this work is important locally, it does not provide any substantial novelty to the current literature.  Saying that, this work still may deserve publication as the results are described appropriately and there is nothing inaccurate about the described work. I have provided several comments with the goal of improving the manuscript. Please see the minor and major comments below for further clarification. 

Major comments:

Title: The title is awkward and not necessarily the object of the manuscript.   Wolbachia is not being introgressed, the mosquito genetic background is being introgressed from one Ae. albopictus strain to another.  Wolbachia is being maternally transferred through the introgression process.  Also, it is not clear how the introgression in enhances ‘locally’.  Do you mean the introgression will enhance an IIT approach that is occurring in a particular location in Spain.  Why not just say this?

The introduction is written as though Wolbachia based IIT and replacement approaches are actually tested and investigated in this paper.  Over half of the introduction material prior to describing the generation of Wolbachia infected lines with a suitable genetic background could be shorted substantially and reference previously published work describing Wolbachia-based IIT and replacement approaches.  

Throughout the manuscript Wolbachia transmission to offspring is referred to as vertical transmission.  While this is not technically incorrect, more often in the literature this is referred to as maternal transmission. It would be more acccurate if vertical transmission is replaced with maternal transmission.

Using a mitochondrial gene is uninformative when trying to compare the genetic diversity of offspring from the introgressions because we know they are going to inherit mitochondrial from their mother. The authors should perhaps consider removing this analysis from the manuscript. At one point, the authors even state the COI was expected not to yield any important information.

It might be worthwhile to discuss why fairly low fecundity and hatch rates (~70%) was observed in the WT and introgressed strains. Is the original ARwP strain fairly inbred?  

There was a small percentage of egg hatch in the incompatible crosses?  Were these just hatched eggs or were larvae also observed?  Were these larvae reared to adults and tested for Wolbachia infection?

Lines 383-386:  I am not sure I agree with the justification provided that Wolbachia negative individuals in ARwPBN and ARwPBA is due to low PCR sensitivity.   If PCR sensitively is so low, can we trust all of your other positive data?

Minor comments:

Lines 42 and 46: Italicize Ae. albopictus

Lines 52-57: These introductory sentences seem arbitrary.  Why not just start this paragraph with line 57…In the current context of global change…

Line 72: remove – Issues like the mosquito. 

Lines 73-74:  Just say effective vaccines have not been developed.

Lines: 95-96:  How common are Wolbachia infections? Just saying very common is too colloquial.  There are multiple meta analyses that can be cited here that suggest estimates of how many insects are infected with Wolbachia.

Lines 112-114:  ­Wolbachia-based IIT is not only currently under testing.  You cited several papers that have demonstrated successful suppression.  Please rework this sentence. 

Lines 122-122: i.e. by crossing wild males with artificially infected females…should be in parentheses. 

Line 136:  Nothing is technically ‘proved’.  Say ‘has been demonstrated’.

Line 243: remover the couple of and replace with ‘the’

Lines 273-274: provide additional details on how eggs were collected.

Line 287: Add ‘An’ before ANOVA

Comments on the Quality of English Language

Please see comments above.

Author Response

RESPONSE TO REVIEWER #1

THE REVIEWER: The authors of this manuscript describe the introgression of the genetic background of an Ae. albopictus population from Spain with a strain of Ae. albopictus harboring a wPip Wolbachia infection. The methods are sound and appropriate and support the authors conclusions. The manuscript is unfortunately full of improper English language usage and needs to be reviewed by a native English speaker and or an editorial service. The statistical analyses appear appropriate where used.  However, the manuscript only described results that are important for the development of an IIT approach to target localized populations in Spain.  While this work is important locally, it does not provide any substantial novelty to the current literature.  Saying that, this work still may deserve publication as the results are described appropriately and there is nothing inaccurate about the described work. I have provided several comments with the goal of improving the manuscript. Please see the minor and major comments below for further clarification. 

ANSWER: We apologize for the improper English usage in the previous version of the manuscript. It has been now fully reviewed and we hope that will make it easier to read.

Major comments:

Title: The title is awkward and not necessarily the object of the manuscript.   Wolbachia is not being introgressed, the mosquito genetic background is being introgressed from one Ae. albopictus strain to another.  Wolbachia is being maternally transferred through the introgression process.  Also, it is not clear how the introgression in enhances ‘locally’.  Do you mean the introgression will enhance an IIT approach that is occurring in a particular location in Spain.  Why not just say this?

ANSWER: The title has been modified to better describe the content of the article: “Wolbachia infection through hybridization to enhance an Incompatible Insect Technique-based suppression of Aedes albopictus in eastern Spain”.

The introduction is written as though Wolbachia based IIT and replacement approaches are actually tested and investigated in this paper.  Over half of the introduction material prior to describing the generation of Wolbachia infected lines with a suitable genetic background could be shorted substantially and reference previously published work describing Wolbachia-based IIT and replacement approaches. 

ANSWER: We agree. Following your advice, we have modified the introduction by removing sentences that were not necessary and eliminating the paragraph describing the Wolbachia-based replacement.

Throughout the manuscript Wolbachia transmission to offspring is referred to as vertical transmission.  While this is not technically incorrect, more often in the literature this is referred to as maternal transmission. It would be more accurate if vertical transmission is replaced with maternal transmission.

ANSWER: We have modified the text following your recommendation.

Using a mitochondrial gene is uninformative when trying to compare the genetic diversity of offspring from the introgressions because we know they are going to inherit mitochondrial from their mother. The authors should perhaps consider removing this analysis from the manuscript. At one point, the authors even state the COI was expected not to yield any important information.

ANSWER: The assay was designed to confirm the introgression of the mtDNA, together with Wolbachia, from ARwP in comparison with the original wild-type line BN that could have a mtDNA carrying different mutations (even if we know that they are rare in the COI among Ae. albopictus populations). For this reason, we would prefer to keep this analysis in the manuscript. Nevertheless, in the discussion we suggest the use of further markers to increase the value of the analysis (page 16, last paragraph).

It might be worthwhile to discuss why fairly low fecundity and hatch rates (~70%) was observed in the WT and introgressed strains. Is the original ARwP strain fairly inbred?

ANSWER: Laboratory settings may induce a significant variation of certain fitness parameters compared to the wild populations and other experiments. Often, a reduced relative humidity can be sufficient to induce a similar effect and we will ascertain it. We have also added the results regarding BN line as Supplementary figure to confirm that the observed results were not associated to lines. Even though inbreeding can be a further causative factor for the observed findings, it is also possible that the BN line was established starting from a small number of individuals.

There was a small percentage of egg hatch in the incompatible crosses?  Were these just hatched eggs or were larvae also observed?  Were these larvae reared to adults and tested for Wolbachia infection?

Larvae were not tested because they should harbor the same Wolbachia infection of their mothers (wAlbA/wAlbB). It is almost impossible that they carry wPip infection or lack of any Wolbachia strain.

Lines 383-386:  I am not sure I agree with the justification provided that Wolbachia negative individuals in ARwPBN and ARwPBA is due to low PCR sensitivity.   If PCR sensitively is so low, can we trust all of your other positive data?

ANSWER: We have revised the lines for clarity (from line 372 on). Our hypothesis is supported by previous findings (see references). The low sensitiveness is related to an extremely low Wolbachia density. False positives are not expected and have not been documented.

Minor comments:

 ANSWER: All minor changes indicated have been taken into account. We will only detail here the comments that need an additional explanation.

Lines 52-57: These introductory sentences seem arbitrary.  Why not just start this paragraph with line 57…In the current context of global change…

ANSWER: We agree in that the indicated introductory sentences are not strictly necessary. However, we do think they may help non-expert readers to contextualize the whole manuscript. Therefore, we would prefer to keep them in the manuscript.

Lines 73-74:  Just say effective vaccines have not been developed.

ANSWER: The sentences have been completely changed (see lines 73-75)

Lines: 95-96:  How common are Wolbachia infections? Just saying very common is too colloquial.  There are multiple meta analyses that can be cited here that suggest estimates of how many insects are infected with Wolbachia.

ANSWER: We thank you for the suggestion. The sentence has been revised and a proper reference has been added (same lines).

Lines 112-114:  ­Wolbachia-based IIT is not only currently under testing.  You cited several papers that have demonstrated successful suppression.  Please rework this sentence. 

 ANSWER: The sentence has been modified as suggested (lines 106-108)

Lines 273-274: provide additional details on how eggs were collected.

ANSWER: The requested details have been added in lines 255-256.

Reviewer 2 Report

Comments and Suggestions for Authors

The research paper focuses on introducing a specific strain of Wolbachia bacteria into the Aedes albopictus mosquito population in Spain to control its population by exploiting the bacteria's ability to cause sterility in the mosquito's offspring. This method is an alternative to traditional insecticides and aims to provide an environmentally friendly solution to managing mosquito populations that carry diseases. The study involves hybridization techniques, genetic analysis, and evaluations of the effectiveness of this control strategy in laboratory settings. The findings suggest potential for field applications and further research into genetically-based mosquito control methods.

I just have a few comments. While the laboratory results are promising, there is a significant leap from laboratory conditions to field applications. I recommend discussing potential challenges and strategies for field implementation, such as the dispersal mechanisms for the Wolbachia-infected mosquitoes and community acceptance. The section on future research directions is insightful. I encourage the authors to elaborate on potential interdisciplinary collaborations that could enhance the scalability and sustainability of this control method, such as partnerships with public health organizations and ecological monitoring groups.

Overall, the manuscript is well-written and structured. Minor revisions for clarity and conciseness in certain sections could further enhance readability.

Author Response

THE REVIEWER: I just have a few comments. While the laboratory results are promising, there is a significant leap from laboratory conditions to field applications. I recommend discussing potential challenges and strategies for field implementation, such as the dispersal mechanisms for the Wolbachia-infected mosquitoes and community acceptance. The section on future research directions is insightful. I encourage the authors to elaborate on potential interdisciplinary collaborations that could enhance the scalability and sustainability of this control method, such as partnerships with public health organizations and ecological monitoring groups.

Overall, the manuscript is well-written and structured. Minor revisions for clarity and conciseness in certain sections could further enhance readability.

ANSWER: We thank the reviewer for the appreciation and for the useful suggestions. We have taken them into account in the revised version of the manuscript, where we have incorporated aspects related to the implementation prospects into the discussion (lines 444-464). We have also worked throughout the whole text to improve clarity and conciseness.

Reviewer 3 Report

Comments and Suggestions for Authors

Overall, I found this is an important and timely study which was well executed (Methods and result interpretation are acceptable. However, it would be better if the fitness parameters of hybrid lines were compared with BN line as well)

However, the manuscript needs revision to publish it in this journal.

In general;

·         I would like to suggest splitting too long sentences for easier readability and better understandability

·         The text is dense with a lot of unnecessary wording and needs rewrite for clarity at some points

·         Consistency throughout the text with using specific terms will enhance understandability.        e.g., the term transinfection was used interchangeably to imply both microinjection (e.g. line 175, 432) and hybridization (e.g. line 235) which would make it confusing for the reader

 Also, following terms are used interchangeably throughout the text; survival/longevity/average     life expectancy, population/ line, egg fertility/egg hatch rate, bidirectional incompatibility/bidirectional CI

·         Consistency throughout the text with using population labels is very important e.g., ARwP or ARwPL, ARwPBN or ARwPBN

Label formats should be similar in the text and figures

·         Any abbreviated term should be spelled out at the first mention with the abbreviation in brackets. e.g., polymerase chain reaction (PCR)

·         Any scientific name should me written fully at the first mention and abbreviated thereafter.    e.g., Aedes albopictus and Ae. albopictus

   Any scientific name should be italicized throughout the text. 

Line by line edits/comments/suggestions (in addition to general flaws);

Line 2 (Title): delete “locally”

                        Spell out IIT

Line 22: “.. target insect species ...”          “... into the environment ...”

Line 44: “Being harmless on non-target organisms and environment, ...”

Line 40: Delete “Despite the starting genetic background....”

Lines 57-61: suggest splitting the sentence

Lines 67-69: could be summarized as    “..... and to develop insecticide resistance”

Lines 72-76: suggest splitting the sentence

Line 87: “....  into lethal gene passed .....”

Line 105: Aedes albopictus

Lines 106-108: good to add a reference

The connection of the fact in lines 105-108 to line 109 -111 could be a little confusing for the reader who is not familiar with the technique. Suggest to rewrite

Line 108: “… that are capable of reducing the vector competence…”

Line 109: “.. associated with the Wolbachia infection, ...”

Line 114: delete “….. together with the number of target areas, …..”

Lines 116-118: re-write for clarity (e.g., “…… generation of a suitable Wolbachia infected laboratory line of the target species of which the males could fully sterilize the wild females.”

Line 120: “... transinfection by microinjection..”

Line 123: may delete “.... because wild males of this species are not infected by Wolbachia and are fully fertile when they inseminate Wolbachia-infected females, ...”

Lines 122-125: suggest splitting the sentence

Line 125: bidirectional CI

Line 127: “…. wild strains (wAlbA and wAlbB) of the species.”

Lines 127-130: suggest splitting the sentence 

Line 129: delete Wolbachia

Line 131:”… laboratory population” or a “ laboratory line” ? Be consistent

Lines 133, 134: delete “…. together with the number of released individuals,”

 Line 134: delete “… the level induced sterility in males should be maximized without affecting the fitness in terms of survival and mating competitiveness ….”

 Line 139: Use “In addition,” for  “At last,”

 Lines 148, 149: Is this sentence correct?

Lines 149-153: a suggestion to change “…. the genetic background of a Wolbachia transinfected Ae. albopictus population into a local wild population of Ae. albopictus in Barcelona, Spain. Results of the characterization of hybrid lines to evaluate their suitability for mass rearing and as a genetic control tool are also reported”.

Line 155: suggest to change Asian Tiger mosquito into Ae. albopictus

Line 156: “... in 2004...”

Lines 158-163: suggest splitting the sentence

Line 167: “The population BN was derived from ….” (be consistent with population or line throughout the text)

Line 168: “…. in Ae. albopictus ….”

Lines 168-170: this population naturally harbors Wolbachia wAlbA and wAlbB. But wild individuals lacking one or both of these two strains have been previously reported in Eastern Spain [57]

Line 171: “…… obtained from the University ….”

Line 172: “…curing the natural Wolbachia infection ….”

Line 175: “... through microinjection of ....”

Lines 175-176: through the transinfection of Wolbachia wPip from Culex pipiens molestus into a wild population which was cured of natural Wolbachia infections ….. …..”

Lines 176-177: “…. bidirectional CI pattern in crosses with wild Ae. albopictus …”

 Line 178: The label ARwPL was not used when describing crosses

Would be good to mention that ARwP (or ARwPL) is the Wolbachia donor in all the experiments in this study

 Line 196, 197: Leave the sub-population labelled as ARwP or label it as ARwPL in subsequent crosses

Line 197:   “ ..... females with 3±1 -day-old BA males.”

Line 197-198: re-write to simplify the sentence

 Delete Lines 207. It is repeated in section 2.5

Line 209: change the positions of BN and BA in the sentence

Line 212: “Vertical inheritance and establishment of the infection

Line 213: replace “establishment with “vertical inheritance”

Lines 215, 217, 229: how did you identify the wPip positive and negative mosquitoes before conducting PCR? Re-write for clarity.

Line 216: replace “vertical inheritance” with “establishment”

Line 217: How these females were tested positive?

Line 217: “…. polymerase chain reaction (PCR)….”

Line 221: Is it Wolbachia infection type or infected Wolbachia strain?

Line 233: :….. cytochrome oxidase I (COI)…..”

starting ones or original populations?

Delete “…. in comparison with the starting ones.”

Line 234: This is repetition “(to ascertain the presence or the successful transinfection of Wolbachia)”

 Line 236: “…… mitochondrial DNA (mtDNA)”

Line 260: Fitness of hybrid lines and induced CI levels”

Line 264: delete: “wild Ae. albopictus

 Line 266: “Specifically, three larval trays were prepared per each population with ….”

 Line 268: Three cages of each population?

Line 269: “Adult survival was measured by counting …”

Line 274: female fecundity (eggs/female) and egg fertility (egg hatch rate)

Line 281: “… standard error (SE),”

Line 286: ”.., egg fertility ….”

Line 287: delete “data”

Line 288: crosses of BN females with males of ARwPBN, ARwPBA, and BN

Line 296: “…average fertility …”

Line 297: delete “on average”

Line 297: I’m confused here. Did you conduct two separate crosses of ARwP females with 3±1 day old and 18±1 day old BN males? If yes, clarify it in the methods and report results on fitness of it as well.

Lines 301-303: “ …., in ARwPBN, the infection by Wolbachia wPip encountered 100% fixation by G3  and continued it up to G5.

 Figure 1: Should not the X-axis title be “Generation number”?

                Use the same label format of ARwPBN and ARwPBA to label the two graphs and in the figure title.

Why G5 is marked by an asterisk?

 Lines 309, 311: The label ARwPL was not used when describing crosses, instead used ARwP

 Section 3.1: In tMaterials and methods it was mentioned that PCR was conducted for the presence/absence of WAlbB and WAlbB. Results?

 Line 315: “mtDNA”

 Line 321: longevity or survival? both terms were used interchangeably in the text

“Fitness and CI levels of hybrid lines compared to ARwP

 Lines 322, 323: “Adult survival” 

Line 323: Figure 2 and Figure 3 are mixed up

Delete “Under the described experimental conditions,”

 Lines 324, 325: survival time / average life expectancy   choose one (appropriate) term to use

 Figure 2: use the same label format as used in the text

What is indicated by error bars?

 Line 331: delete Ae. albopictus

 Line 335: delete Ae. albopictus

 Line 336: check Fig no.

 Line 338: “Immature survival of…”

Delete Ae. albopictus.

 Line 343: delete (p set at <0.05)

 Line 344: Delete “Accordingly, ..”

 Line 348: egg fertility or egg hatch rate? (be consistent with the term used)

Delete Ae. albopictus (or be consistent with its use throughout the text)

 Line 353: spell out SEM

 Line 354: delete (p set at <0.05)

Line 355: “…. Males have acquired an almost full level ….”

Line 356:  “… and the differences in egg fertility compared to the control were significant …”

What are the post hoc  P values for two hybrid lines?

 Lines 356, 357: differences compared to BN males were clearly significant INTO differences compared to the control were significant

 Line 357: delete clearly, check the label of BN, check the P value P < = 0.05,

“The difference …..”

 Line 360: egg fertility or egg hatch rate? Delete Ae. albopictus

 Line 361: “…. with males of …”

 Line 365: check the sentence of accuracy

 Line 366: delete (p set at <0.05)

 Line 393: delete “in BN females”

Lines 396-397: contradictory as there was no evident fitness effects

Line 428: by introducing a foreign Wolbachia strain.

Line 438: transinfection into infection

Line 439: moving into injecting

Line 447-455: hard to understand. rewrite

Line 368 (Discussion): More emphasis on the successful transfer of wPip by G3 without any fitness cost on the population would enhance the quality of the manuscript

Also, discuss any possible reasons of the observed reduction of wPip transferring after G2 in ARwPBA

It would be better if the fitness parameters of hybrid lines were compared with BN line as well

Line 384: “… Positive for the…..”

LIne 396: “ ….. tested Ae. albopictus lines …”

Line 397: Tested fitness effects were not significantly different. What are the other evident fitness effects?

Line 402: “…over time, due to selection and ……”

Lines 401-404: suggest splitting the sentence for clarity

Line 426: “….. in an area of eastern Spain. Both lines were capable of fully sterilizing…”

Line 428: delete “ …. thanks to the introduction of a foreign Wolbachia infection.”Line 438: delete “whenever possible”

Line 441: “... ideal characteristics ...”

Lines 461-468: Delete

Comments on the Quality of English Language

Overall, the English language usage in the manuscript is acceptable. However, some sentences are too long making it hard to read and understand. I suggest rewriting of some sentences/sections for better readability and understandability

Author Response

REVIEWER 3: Overall, I found this is an important and timely study which was well executed (Methods and result interpretation are acceptable. However, it would be better if the fitness parameters of hybrid lines were compared with BN line as well).

ANSWER: First of all, we want to thank the reviewer for the detailed report that we have found very useful to improve the manuscript.

               As for the specific question about the comparison of the fitness parameters with the BN line, we want to point out that it was not a main objective of the work because we intended to evaluate the effect of wPip Wolbachia on different genetic backgrounds. However, we understand the point of view of the Reviewer. We do have data regarding female fecundity and female fertility of the BN line, which did not reveal any differences with the Wolbachia-infected lines and we have added them as a Supplementary figure (Figure S2).

THE REVIEWER: However, the manuscript needs revision to publish it in this journal. In general;

  • I would like to suggest splitting too long sentences for easier readability and better understandability
  • The text is dense with a lot of unnecessary wording and needs rewrite for clarity at some points
  • Consistency throughout the text with using specific terms will enhance understandability.        e.g., the term transinfection was used interchangeably to imply both microinjection (e.g. line 175, 432) and hybridization (e.g. line 235) which would make it confusing for the reader

 Also, following terms are used interchangeably throughout the text; survival/longevity/average     life expectancy, population/ line, egg fertility/egg hatch rate, bidirectional incompatibility/bidirectional CI

  • Consistency throughout the text with using population labels is very important e.g., ARwP or ARwPL, ARwPBN or ARwPBN. Label formats should be similar in the text and figures
  • Any abbreviated term should be spelled out at the first mention with the abbreviation in brackets. e.g., polymerase chain reaction (PCR)
  • Any scientific name should me written fully at the first mention and abbreviated thereafter.    e.g., Aedes albopictus and Ae. albopictus

   Any scientific name should be italicized throughout the text. 

ANSWER: The manuscript has been revised according to all of the reviewer’s comments and suggestions, and also to improve the English usage. Thus, overly long sentences have been now split and simplified; we have revised the manuscript to improve consistency in the use of specific terms with regard to Wolbachia infection approaches, fitness parameters, and labels; we have fully written all the abbreviated terms and species names when first mentioned, and checked the scientific names.

               Below we respond to certain specific requests that need to be further detailed.

Lines 106-108: good to add a reference

ANSWER:  The reference by Zug and Hammerstein (2012) has been added as number 26.

The connection of the fact in lines 105-108 to line 109 -111 could be a little confusing for the reader who is not familiar with the technique. Suggest to rewrite.

ANSWER: To avoid confusion and since this part is not strictly related to IIT, this part has been eliminated.

Line 131:”… laboratory population” or a “ laboratory line” ? Be consistent capable of reducing

ANSWER: To avoid confusions, in this revised version of the manuscript we use the term “population” only when referring to wild individuals, while we use “line” to refer to all laboratory mosquitoes not having, including colonized populations, Wolbachia cured populations and hybrid populations.

 Line 134: delete “… the level induced sterility in males should be maximized without affecting the fitness in terms of survival and mating competitiveness ….”

ANSWER: The whole paragraph been revised for clarity. See Lines 124-126

Lines 148, 149: Is this sentence correct?

Lines 149-153: a suggestion to change “…. the genetic background of a Wolbachia transinfected Ae. albopictus population into a local wild population of Ae. albopictus in Barcelona, Spain.

ANSWER: These sentences have been modified to improve clarity. See Lines 139-141

Line 178: The label ARwPL was not used when describing crosses

Would be good to mention that ARwP (or ARwPL) is the Wolbachia donor in all the experiments in this study Line 196, 197: Leave the sub-population labelled as ARwP or label it as ARwPL in subsequent crosses

ANSWER: The use of the labels has been revised throughout the text.

Delete Lines 207. It is repeated in section 2.5.

ANSWER: In section 2.2 we talk about the egg fertility in the crosses of ARwPL females × aged BN males measured each generation. In section 2.5 we refer only to the G5 measurements for the crosses of ARwPBA and ARwPBN males with BN females.

Line 212: “Vertical inheritance and establishment of the infection

ANSWER: The title of this subsection has been modified based on the comment regarding line 216. Now it is “Monitoring of the establishment and maternal transmission of the infection “

Line 213: replace “establishment with “vertical inheritance”

ANSWER: Please, see the answer to the comment about line 216

Lines 215, 217, 229: how did you identify the wPip positive and negative mosquitoes before conducting PCR? Re-write for clarity.

ANSWER: This part has been rewritten to improve clarity. See lines 206-210.

Line 216: replace “vertical inheritance” with “establishment”

ANSWER: The vertical inheritance was evaluated by analyzing the percentage of positive daughters born from a positive female only at G5. Instead, individuals from G1 to G4 were sampled randomly, without knowing whether their mother was positive or not, just to evaluate the gradual establishment of the infection. The individuals showing a perfect vertical inheritance are supposed to increase generation by generation through selection. For this reason, instead of the proposed change, we modified the text to make it clear.

Line 217: How these females were tested positive?

ANSWER: They were tested by PCR. The sentence has been revised for clarity (lines 206-210).

Line 221: Is it Wolbachia infection type or infected Wolbachia strain?

ANSWER: The whole sentence has been removed in this new version of the manuscript.

Line 233: starting ones or original populations?

Delete “…. in comparison with the starting ones.”

ANSWER: The title of the paragraph has been revised by eliminating these words

Line 260: Fitness of hybrid lines and induced CI levels”

ANSWER: The sentence has been revised accordingly

Line 297: I’m confused here. Did you conduct two separate crosses of ARwP females with 3±1 day old and 18±1 day old BN males? If yes, clarify it in the methods and report results on fitness of it as well.

ANSWER: We have described two protocols of hybridization in the methods section. BA males were used when they were 3 days old (lines 189-194), while BN males were used when they are 18 days old to exploit the decrease in the CI level due to the depletion of wAlbA (lines 195-200).

 Figure 1: Should not the X-axis title be “Generation number”? Use the same label format of ARwPBN and ARwPBA to label the two graphs and in the figure title. Why G5 is marked by an asterisk?

ANSWER: Figure 1 has been modified as suggested. The asterisk indicated that a different method was used to measure the maternal transmission of wPip (as indicated in subsection 2.3, 10 females sampled randomly from G1 to G4, 20 females born to mothers previously tested positive for wPip in G5). The asterisk has now been removed as this point is also highlighted with a darker color of the column.

 Section 3.1: In Materials and methods it was mentioned that PCR was conducted for the presence/absence of WAlbB and WAlbB. Results?

ANSWER: We have added this information in the Results section (lines 287-288).

 Line 321: longevity or survival? both terms were used interchangeably in the text.

Lines 324, 325: survival time / average life expectancy   choose one (appropriate) term to use

ANSWER: It has been revised and now only the word “survival” is used throughout the text.

Figure 2: use the same label format as used in the text

What is indicated by error bars?

ANSWER: The labels have been modified and the information about the error bars has been added to the legend.

 Line 348: egg fertility or egg hatch rate? (be consistent with the term used).  

Delete Ae. albopictus (or be consistent with its use throughout the text)

ANSWER: It has been revised and now only “egg fertility” is used throughout the text. We have also checked the way of mentioning the hybrid lines throughout the text.

Line 356:  What are the post hoc  P values for two hybrid lines?

ANSWER: They show that the level of CI with wild females is not significantly different between the two hybrid lines.

Lines 396-397: contradictory as there was no evident fitness effects

ANSWER: The sentence about the fitness cost of Wolbachia wPip in Ae. albopictus refers to the first documented introduction by Calvitti et al (2010), reference 32. 

Line 447-455: hard to understand. Rewrite

ANSWER: We have fully revised this section. The new paragraph does from lines 436-443.

Line 368 (Discussion): More emphasis on the successful transfer of wPip by G3 without any fitness cost on the population would enhance the quality of the manuscript

ANSWER: Thank you for the suggestion. We agree with the importance of the fast achievement of the successful infection. However, we do consider 5 generations as a minimum for an outcrossing protocol as, at G3, hybrids still had a significant proportion of ARwPL genomic background instead of that from and not BN genes (even if these differences could be not relevant).

Also, discuss any possible reasons of the observed reduction of wPip transferring after G2 in ARwPBA we

ANSWER: We have already discussed these data by referring to the possibility of a gradual counter-selection of  females with not perfect maternal transmission (Lines 372-383). At G5, the CI level determined by ARwPBA males was 99.9% therefore we expect the full fixation of the infection to occur.

It would be better if the fitness parameters of hybrid lines were compared with BN line as well

ANSWER: we have added data regarding BN related to female fecundity and egg fertility in S2 figure

Line 397: Tested fitness effects were not significantly different. What are the other evident fitness effects? ANSWER: The sentence has been revised for clarity

Round 2

Reviewer 1 Report

Comments and Suggestions for Authors

Lines 202-204.  This sentence should read …embryonic microinjection of wPip isolated from a wild population of Culex pipiens molestus.  

Line 231-232: Change to the…The sex was determined of the progeny to allow for the use….

Figures 1-5: The image quality of these figures is low.  Perhaps, this is  function of the submission system and how a final pdf is generated for review.   Please improve image quality before acceptance. 

Comments on the Quality of English Language

None

Author Response

We thank  again Reviewer1 for the valuable suggestions.

The manuscript has been revised accordingly.

Specifically,

the sentence at lines 202-204 has been integrated with "...wPip isolated from a wild population..."

the sentence at lines 231-232 has been revised with the words that have been suggested.

Figures have been checked for quality and a higher quality version has been also included in the manuscript.

Best regards